# Effect of Hypoxia on Pulmonary Endothelial Cells from Bleomycin-Induced Pulmonary Fibrosis Model Mice

**DOI:** 10.3390/ijms23168996

**Published:** 2022-08-12

**Authors:** Daisuke Akahori, Naoki Inui, Yusuke Inoue, Hideki Yasui, Hironao Hozumi, Yuzo Suzuki, Masato Karayama, Kazuki Furuhashi, Noriyuki Enomoto, Tomoyuki Fujisawa, Takafumi Suda

**Affiliations:** 1Second Division, Department of Internal Medicine, Hamamatsu University School of Medicine, 1-20-1 Handayama, Higashi-ku, Hamamatsu 431-3192, Shizuoka, Japan; 2Department of Clinical Pharmacology and Therapeutics, Hamamatsu University School of Medicine, 1-20-1 Handayama, Higashi-ku, Hamamatsu 431-3192, Shizuoka, Japan

**Keywords:** α-SMA, bleomycin, endothelial cell, hypoxia, HIF-α, pulmonary fibrosis

## Abstract

Pulmonary fibrosis is a progressive and fatal disorder characterized by dysregulated repair after recurrent injury. Destruction of the lung architecture with excess extracellular matrix deposition induces respiratory failure with hypoxia and progressive dyspnea. The impact of hypoxia on pulmonary endothelial cells during pulmonary fibrogenesis is unclear. Using a magnetic-activated cell sorting system, pulmonary endothelial cells were isolated from a mouse model of pulmonary fibrosis induced by intratracheally administered bleomycin. When endothelial cells were exposed to hypoxic conditions, a hypoxia-inducible factor (HIF)-2α protein was detected in CD31- and α-smooth muscle actin (SMA)-positive cells. Levels of plasminogen activator inhibitor 1, von Willebrand factor, and matrix metalloproteinase 12 were increased in endothelial cells isolated from bleomycin-treated mice exposed to hypoxic conditions. When endothelial cells were cultured under hypoxic conditions, levels of fibrotic mediators, transforming growth factor-β and connective tissue growth factor, were elevated only in endothelial cells from bleomycin-treated and not from saline-treated lungs. The increased expression of α-SMA and mesenchymal markers and collagen production in bleomycin- or hypoxia-stimulated endothelial cells were further elevated in endothelial cells from bleomycin-treated mouse lungs cultured under hypoxic conditions. Exposure to hypoxia damaged endothelial cells and enhanced fibrogenesis-related damage in bleomycin-treated pulmonary endothelial cells.

## 1. Introduction

Interstitial lung diseases and the advanced and end-stage pathological condition, pulmonary fibrosis, are a diverse group of respiratory diseases that are characterized by various patterns of interstitial inflammation, cell proliferation, and fibrillization in the lung parenchyma [1]. Although the underlying pathogenesis remains to be fully elucidated, recurrent epithelial cell injury and subsequent dysregulated repair are considered fundamental processes [2,3,4,5]. Injury mainly occurs within the alveolus and the interstitium, in the space between the epithelial and endothelial basement membranes [1,6]. Uncontrolled repair causes the accumulation of excess extracellular matrix.

In the lung, complex interactions among epithelial cells, alveolar macrophages, and inflammatory and fibrogenic effector cells are assumed to contribute to fibrotic changes [7,8]. Activated and unremoved fibroblasts secrete collagen and profibrotic mediators and transdifferentiate into myofibroblasts [9]. Myofibroblasts are fibroblast-like cells expressing α–smooth muscle actin (SMA) [2,7,10,11] and are primary effector cells of the fibrotic response and tissue remodeling [7,12]. During repair and reconstruction, myofibroblasts produce large amounts of collagen and other extracellular matrix components.

Endothelial cells are involved in diverse functions, including vasomotor activity, permeability, angiogenesis, trafficking of leukocytes, and innate and acquired immunity. Endothelial cells have structural and phenotypic heterogeneity, which are derived from the surrounding microenvironment [13,14]. Pulmonary endothelial cells configure as a monolayer on the lung intravascular surface and are anatomically contiguous with epithelial and mesenchymal cells [14]; therefore, the pulmonary endothelium can play an important role in the fibrotic response together with epithelial, inflammatory and fibrogenic effector cells [6,7,8,15,16]. Previously, we characterized pulmonary endothelial cells isolated from an in vivo mouse model of pulmonary fibrosis induced by intratracheally administered bleomycin and reported changes in the properties in pulmonary endothelial cells during bleomycin-induced fibrogenesis [17]. Expression levels of endothelial injury markers were elevated in endothelial cells from bleomycin-treated lungs and intracellular nitric oxide production in response to thapsigargin stimulation was significantly lower in endothelial cells from bleomycin-treated mice. Furthermore, endothelial cells from bleomycin-treated mouse lungs had increased expression of mesenchymal markers, which indicated an endothelial to mesenchymal transition as a source of myofibroblasts during fibrogenesis [5,12,18].

Patients with interstitial lung diseases and pulmonary fibrosis have diverse and unpredictable clinical courses, some of which are progressive and fatal [1,3,4,7]. Alveolar septal thickening and destruction of the lung architecture with excess accumulation of extracellular matrix often cause impaired alveolar diffusion and reduced gas exchange, which can induce respiratory failure with hypoxia and impaired tissue oxygenation [1,2,19,20]. Hypoxia is a prominent feature of pulmonary fibrosis [19], and worsening hypoxia correlates with fibrotic disease progression [21]. Adequate oxygenation is essential and an inadequate oxygen supply can lead to cell, tissue, organ and, ultimately, organism death; therefore, complex mechanisms to avoid hypoxia have evolved [22,23]. When hypoxia occurs, oxygen-sensing mechanisms help the body adapt to hypoxia. One of these mechanisms is the transcriptional response to hypoxia, which is mainly mediated by the expression of hypoxia-inducible transcription factors (HIFs), inducible oxygen-sensitive α subunits, HIF-1α, HIF-2α and HIF-3α, and a constitutively expressed β subunit, HIF-β [22,23,24]. The expression and function of HIF-α are strictly regulated, and HIF-α is inactive when oxygen is abundant. In normoxic conditions, HIF-α is rapidly ubiquitinated and subjected to proteasomal degradation [22,23,24,25]. Under hypoxic conditions, HIF-α is not ubiquitinated and forms a functional transcriptional heterodimer with the HIF-β subunit, which promotes transcription of numerous target genes that have a hypoxia-response element in their promoter or enhancer. This regulates expression of these genes to maintain oxygen homeostasis. 

We previously showed that pulmonary endothelial cells isolated from an in vivo bleomycin-induced pulmonary fibrosis model had altered properties related to fibrogenesis [17], but the impact of hypoxia and the expression of HIFs in pulmonary endothelial cells are unclear. It remains unknown whether hypoxia or HIF activation promotes endothelial to mesenchymal transition in pulmonary endothelial cells. In this study, we examined the responses of endothelial cells isolated from bleomycin-induced pulmonary fibrosis model mice to hypoxic conditions.

## 2. Results

### 2.1. Intratracheal Administration of Bleomycin Induced Inflammatory Cell Infiltration and Pulmonary Fibrosis

Intratracheal administration of bleomycin to mice using a microsprayer aerosolizer led to pathological changes with patchy inflammatory cell infiltration and epithelial injury with reactive hyperplasia (Appendix A). Masson’s trichrome staining showed deposits of collagen fibers in bleomycin-treated lungs. Bleomycin induces time-dependent morphological injury, interstitial inflammation, and deposition of extracellular matrix [26]. Inflammatory cell infiltration in the alveolar walls was marked at day 7, indicating prominent inflammation. We isolated pulmonary endothelial cells 7 days after bleomycin or saline administration.

### 2.2. Expression of HIF-α in Pulmonary Endothelial Cells

We cultured endothelial cells from saline or bleomycin-treated mouse lungs under normoxic and hypoxic (1% oxygen) conditions and evaluated the levels of HIF-α protein and mRNA. Under normoxic conditions, HIF-1α and HIF-2α proteins were not detected in endothelial or 3LL cells. When cells were exposed to hypoxic conditions for 72 h, both HIF-1α and HIF-2α proteins were detected in 3LL cells (Figure 1A). HIF-2α protein was detected in endothelial cells from saline- and bleomycin-treated mouse lungs cultured under hypoxic conditions. However, HIF-1α protein was not detected in endothelial cells from either saline- or bleomycin-treated mouse lungs. Upregulation of HIF-α mRNA was not detected in endothelial cells even under hypoxic conditions (data not shown). When endothelial cells were fluorescently stained for HIF-2α and CD31, HIF-2α was detected in CD31-positive cells from saline- and bleomycin-treated mouse lungs cultured under hypoxic conditions (Figure 1B). The percentage of HIF-2α positively stained cells was 1.13% under hypoxic conditions, which was 4.5 times higher than that under normoxic conditions (0.25%, *p* < 0.05).

### 2.3. Pulmonary Endothelial Cells Are Injured by Bleomycin Administration and Exposure to Hypoxia 

Levels of plasminogen activator inhibitor 1 (PAI-1), von Willebrand factor (vWF), and matrix metalloproteinase (MMP)-12, were evaluated as endothelial injury markers (Table 1). In endothelial cells from bleomycin-treated lungs, mRNA levels of these injury markers were significantly elevated. When endothelial cells from saline and bleomycin-treated mice were cultured under hypoxic conditions, PAI-1, vWF and MMP-12 mRNA levels were upregulated. vWF and MMP-12 mRNA levels were significantly increased in endothelial cells from bleomycin-treated mice compared with those from saline-treated mice under hypoxic conditions (*p* < 0.05). Therefore, expression of these endothelial injury markers was most markedly increased when endothelial cells from bleomycin-treated lungs were cultured under hypoxic conditions. These data indicate that exposure to hypoxic conditions damaged endothelial cells and that pre-stimulation with bleomycin enhanced hypoxia-induced endothelial cell injury.

### 2.4. Expression of Fibrotic Mediators and NOSs in Endothelial Cells Exposed to Hypoxic Conditions

We examined mRNA levels of fibrotic mediators (Figure 2). In endothelial cells from bleomycin-treated mouse lungs, mRNA levels of transforming growth factor beta 1 (TGF-β1), connective tissue growth factor (CTGF), and platelet-derived growth factor (PDGF) family members, PDGF-B, PDGF-C, and PDGF-D, were significantly elevated compared with those in endothelial cells from saline-treated mice. In endothelial cells from bleomycin-treated mice exposed to low oxygen concentrations, expression of TGF-β1, CTGF, and PDGFs was further elevated. The increased levels of these mediators were largest in endothelial cells from bleomycin-stimulated lung cultured under hypoxic conditions. When endothelial cells from saline-treated lungs were cultured under 1% oxygen, the expression of TGF-β1 and CTGF was not increased. Although levels of the CTGF protein in the culture medium were elevated for endothelial cells from bleomycin-treated mice and for those exposed to hypoxic conditions, culture under hypoxic conditions did not further increase TGF-β1 protein levels (Figure 3). The collagen content released into the culture medium was elevated for endothelial preparations from bleomycin-treated mice that were cultured under hypoxic conditions (Figure 4). Hypoxic conditions elevated endothelial nitric oxide synthase (eNOS) and inducible nitric oxide synthase (iNOS) expression in endothelial cells from saline and bleomycin-treated mice (Figure 2). iNOS expression was most increased when endothelial cells from bleomycin-treated lung were cultured under 1% oxygen. 

### 2.5. Phenotypic Changes in Endothelial Cells Cultured under Hypoxic Conditions 

Next, we explored the phenotypic changes of endothelial cells from bleomycin-treated lungs cultured under hypoxic conditions using immunofluorescence analysis. We fluorescently stained endothelial cells for α-SMA and CD31 (Figure 5). The percentage of α-SMA-positive cells among endothelial cells from bleomycin-treated lungs was 6.5% under normoxic and 21.2% under hypoxic conditions, which was a statistically significant difference (*p* < 0.01). Endothelial cells from saline-treated lungs cultured under 1% hypoxic conditions had a higher percentage of α-SMA-positive cells (5.6% in 1.0% oxygen, *p* < 0.05). In addition, to examine whether hypoxia accelerated bleomycin-induced endothelial to mesenchymal transition, we examined the expression of the transcription factors, Twist-1, Snail, and Slug, in endothelial cells cultured under hypoxic conditions. These transcription factors participate in regulating the gene expression involved in the induction of endothelial to mesenchymal transition [27]. mRNA levels of mesenchymal markers in bleomycin-stimulated lung endothelial cells were increased when exposed to hypoxic conditions (Figure 6). Expression of mesenchymal marker Snail in lung endothelial cells, already increased by bleomycin treatment, was further increased by exposure to hypoxic conditions. HIF-2α fluorescence signal was detected both in CD31-positive cells and α-SMA-positive cells (Figure 7).

## 3. Discussion

Damage and dysregulated repair of lung architecture lead to impaired alveolar gas exchange and tissue hypoxia. Hypoxia is a main characteristic of pulmonary fibrosis and patients with interstitial lung diseases and pulmonary fibrosis suffer from varying degrees of dyspnea. In this study, we examined the effects of hypoxia on endothelial cells from an in vivo mouse model of pulmonary fibrosis induced by intratracheally administered bleomycin. In endothelial cells cultured under 1% oxygen, levels of endothelium injury markers and HIF-2α were elevated. Expression of mediators related to fibrosis and collagen production was also increased. Expression of mesenchymal markers was increased in endothelial cells exposed to hypoxic conditions compared with that in cells under normoxic conditions. Hypoxia induced additional changes in bleomycin-stimulated lung endothelial cells, potentially related to fibrogenesis.

We confirmed that stimulation with bleomycin increased the expression of fibrotic mediators in endothelial cells. Epithelial cells, alveolar macrophages, immune cells, and fibroblasts contribute to the development of tissue fibrosis by producing various mediators, such as CTGF [28], TGF-β [5,6], and PDGF [29], and endothelial cells also produce such agents and participate in the pathogenesis of fibrosis. Culture of endothelial cells from bleomycin-induced pulmonary fibrosis model mice in hypoxic conditions further elevated the mRNA levels of these fibrotic mediators, but this was not the case for endothelial cells from saline-treated mouse lungs. The magnitude of this upregulation was largest when bleomycin-stimulated lung endothelial cells were cultured under 1% oxygen. In addition, exposure to hypoxic conditions increased CTGF proteins in endothelial cells from bleomycin-induced pulmonary fibrosis. Bleomycin-stimulation might represent a priming state of fibrosis, causing the production of more fibrotic mediators and collagen, and any kind of additional stimulation, hypoxia in our case, may drive the progression of fibrosis.

The ability to respond to changes in oxygen concentration is a fundamental requirement for the survival of all organisms. HIFs are master regulators of adaptation to hypoxia [22,23,30,31], and can be involved in disease pathogenesis [23,24]. HIF-α has two key isoforms, HIF-1α and the highly homologous HIF-2α. HIF-1α exhibits a ubiquitous expression pattern and is important in oxygen homeostasis. HIF-2α is found in a limited number of cell types, including endothelial cells, cardiomyocytes, hepatocytes, glial cells, and interstitial cells of the kidney [23,32,33]. In this study, culture under 1% oxygen for 72 h increased HIF-2α protein levels in endothelial cells from both saline- and bleomycin-treated mouse lungs, which indicated that endothelial cells have the ability to respond to hypoxic stimulation.

It is possible that the threshold of exposure to hypoxia to induce expression was different between HIF-1α and HIF-2α. In idiopathic pulmonary fibrosis (IPF) fibroblasts, 3% hypoxia induced different expression time courses for HIF-1α and HIF-2α [34]. HIF-1α levels peaked within 24 h of hypoxia exposure while HIF-2α protein levels progressively increased over 72 h. Bartoszewski et al. reported the time course of HIF-1α and HIF-2α expression during hypoxia in 10 different primary human endothelial cell lines from different vascular beds. HIF-1α rapidly accumulated during acute hypoxia and its abundance was gradually reduced in response to chronic hypoxia. HIF-2α slowly decreased and remained decreased for longer [33]. Franke et al. showed that HIF-2α is the predominant HIF subunit after chronic exposure to hypoxia [35]. In our study, HIF-1α protein was not detected in endothelial cells cultured under hypoxic conditions for 24 or 72 h (data not shown). Induction of HIF expression might depend on the anatomical origin of cells. Stroka et al. examined the spatial expression of HIF in mice subjected to varying oxygen concentration and showed that expression of HIF-α was organ-dependent. When mice were exposed to 6% oxygen, HIF-1α was detected by immunoblotting in the brain, kidney, liver, spleen, and heart, but not in lung tissue [36]. In an alveolar macrophage cell line cultured under <1% oxygen, the HIF-1α protein level was increased [37]. Higgins et al. examined the role of HIF-1 in the development of mouse renal fibrosis and showed that HIF-1α enhanced epithelial to mesenchymal transition (EMT) in vitro [38]. The loss of epithelial HIF-1α decreased inflammatory cell infiltration and interstitial collagen deposition, and inhibited the development of tubulointerstitial fibrosis. Bryant et al. showed that HIF-1α and HIF-2α proteins were upregulated when lung endothelial cells from bleomycin-intraperitoneally-injected mice were exposed for 6 h to 1% oxygen conditions [39]. Lung tissues and cells are directly exposed to various atmospheric oxygen conditions; therefore, there may be a different mechanism for the induction of HIFs in endothelial cells located in airways.

HIF-1α plays critical roles in fibrotic progression in various organs [37,38,40,41]. Tzouvelekis et al. examined the pathological role of HIF-1α in a bleomycin-induced fibrosis mouse model and in patients with IPF and organizing pneumonia [41]. mRNA levels of HIF-1α were increased in lung tissues and immunohistochemistry showed overexpression of HIF-1α mainly at the epithelium of bleomycin-induced fibrotic mice. Unfortunately, however, Tzouvelekis et al. did not examine HIF-2α. The importance of HIF-2α in pulmonary fibroblast cells has been reported. Knockdown of HIF-2α decreased hypoxia-induced fibroblast proliferation, which indicated the importance of HIF-2α in fibroblast proliferation [34]. Senavirathna et al. investigated the effect of hypoxia on pulmonary fibroblast proliferation. Hypoxia for 3–6 days increased the proliferation of fibroblasts from normal subjects and IPF patients. Inhibition of HIF-2α using an HIF-2-specific inhibitor reduced human pulmonary fibroblast proliferation in hypoxic conditions, indicating that hypoxia can induce the proliferation of human pulmonary fibroblasts through HIF-2α [19]. Jiang et al. used the tracheal microvasculature of mice to show that HIF-2α but not HIF-1α plays an essential role in endothelial cells in maintaining the structure and function of airway microvessels [30]. Genetic deletion of HIF-2α caused apoptosis of tracheal endothelial cells, diminished pericyte coverage, reduced vascular perfusion, impaired barrier function, overlying epithelial abnormalities, and subepithelial fibrotic remodeling. HIF-2α may be more crucial than HIF-1α in endothelial cells under physiologically hypoxic conditions and is a promising therapeutic target. Most HIF-regulated genes are regulated by both HIF-1α and HIF-2α, while some are regulated distinctly by HIF-1α and HIF-2α [33,42]. Although the reason for the discrepancy in the involvement of HIF-2α or HIF-1α remains unclear, two different HIF-α subunits can achieve a more flexible oxygen response system [32]. It is reasonable that in endothelial cells both HIF-1 and HIF-2 regulate the response to adaptation by activating signaling cascades that promote endothelial migration, growth, and differentiation.

Hypoxia can induce the differentiation of fibroblasts into myofibroblasts. To determine the role of fibroblast HIF-α in pulmonary fibrosis, Goodwin et al. examined fibroblasts derived from bleomycin-induced pulmonary fibrosis animal models and showed that hypoxia significantly enhanced TGF-β-induced myofibroblast differentiation and α-SMA expression through HIF-1α [43]. They did not target HIF-2α signaling in fibroblasts. HIF-1α knockdown significantly reduced TGF-β-induced α-SMA in normoxic and hypoxic conditions. It remains unknown whether hypoxia can induce endothelial-mesenchymal transition. In this study, we focused on the effect of hypoxia on endothelial cells isolated from lung tissues with bleomycin-induced fibrosis and investigated the phenotypical changes occurring during fibrogenesis and hypoxia. Hypoxia increased the expression of α-SMA-positive endothelial cells from bleomycin-induced pulmonary fibrosis mice. HIF-2α expression was confirmed both in CD31- and α-SMA-positive cells. The increase in α-SMA levels in response to hypoxia was more marked in endothelial cells from bleomycin-induced fibrotic lung tissues, indicating that pre-stimulation with bleomycin sensitized endothelial cells to hypoxic stress and accelerated hypoxia-induced endothelial to mesenchymal transition. Using pulmonary microvascular endothelial cells from rats with hypoxia-induced pulmonary hypertension, Zhang et al. showed that incubation of these cells in 1% oxygen induced a transition from endothelial to smooth muscle-like cells, which indicated endothelial to mesenchymal transition [44].

In our study, the collagen content was elevated in endothelial cells from bleomycin-treated mouse lung. Interestingly, culture under hypoxic conditions increased the collagen content even in saline-treated and bleomycin-treated pulmonary endothelial cells to comparable levels. In addition, low oxygen conditions increased the number of α-SMA-positive cells even in endothelial cells from saline-treated mouse lungs. Hypoxia itself may have critical roles in the pathogenesis of pulmonary fibrosis and accelerate its progression. Hypoxia itself is proposed to be an important microenvironmental factor in the pathogenesis of pulmonary fibrosis [38]. Oxygen therapy is widely applied in patients with pulmonary fibrosis and maintaining sufficient oxygen status may suppress progression of fibrosis. 

Endothelial cells have the capacity to produce nitric oxide and the production of nitric oxide is regulated by the expression of NOS [45]. Nitric oxide protects cells from oxidant-induced injury [46] and knockout of eNOS induces prolonged fibrosis after bleomycin exposure [47], indicating that endothelial cells may be involved in pulmonary fibrosis by production of nitric oxide. In this study, endothelial cells from bleomycin-induced fibrosis showed increased eNOS and iNOS expression in response to hypoxia. Although hypoxia induced the expression of iNOS and eNOS in endothelial cells from saline-treated control mice, it was less than that in bleomycin-treated endothelial cells. In response to varying oxygenation, nitric oxide levels are modified in endothelial cells by differential expression of the NOS and HIF isoforms. eNOS mainly controls nitric oxide production under normoxic conditions [48], while iNOS mainly controls nitric oxide synthesis under hypoxic conditions [49]. In this study, endothelial cells from bleomycin-treated mice showed significantly increased iNOS and eNOS mRNA levels after hypoxic culture. As for the relationship between HIFs and nitric oxide, Branco-Price et al. showed that hypoxic conditions did not change nitric oxide synthesis in endothelial cells without HIF-1α [48].

In conclusion, we focused on the effect of hypoxia on endothelial cells from bleomycin-induced pulmonary fibrosis. Hypoxia accelerated the fibrogenic changes of bleomycin-induced endothelial cells, which were already functionally altered with a phenotype similar to myofibroblasts and producing fibrogenic mediators and undergoing endothelial to mesenchymal transition. A pathway involving endothelial cells and HIFs may be a new therapeutic target for pulmonary fibrosis.

## 4. Materials and Methods

### 4.1. Bleomycin-Induced Pulmonary Fibrosis Mouse Model

C57/BL6 male mice 9 to 12 weeks old (20–25 g, SLC, Shizuoka, Japan) were anesthetized with intraperitoneal ketamine (80 mg/kg) and xylazine (10 mg/kg). Using a microsprayer (Microsprayer Aerosolizer; PennCentury, Philadelphia, PA, USA), mice were instilled a single tracheal injection of 2 mg/kg bleomycin sulfate (Nippon Kayaku, Tokyo, Japan) in 50 μL sterile saline on day 0. Control mice received tracheal injection of 50 μL sterile saline. All mice were killed by cervical dislocation and lungs were harvested at day 7 after intratracheal saline or bleomycin injection. This study was approved by the Animal Care and Use Committee of Hamamatsu University School of Medicine and all experiments were performed according to guidelines of this Committee.

### 4.2. Histopathology

The lungs of mice were removed and inflated using a syringe. Then, the specimens were fixed in 10% buffered formalin and embedded in paraffin. Four µm of sections cut from embedded tissues were stained with hematoxylin–eosin (HE) and Masson’s trichrome stains and observed under a light microscope.

### 4.3. Isolation of Mouse Pulmonary Endothelial Cells

Mouse pulmonary endothelial cells were isolated as previously described [17]. Briefly, after harvesting, lungs were minced with sterile scissors and incubated with 100 U/mL DNase 1 (Worthington, Lakewood, NJ, USA) and 200 U/mL collagenase type 2 (Worthington, Lakewood) for 40 min at 37 °C in phosphate-buffered saline (PBS). The tissue samples were then gently dissociated into single cell suspensions using a MACS Dissociator (Myltenyi Biotechnology, Bergisch Gladbach, Germany), according to the manufacturer’s protocol. CD45 microbeads (Myltenyi Biotec) were then incubated with the single cell suspensions to achieve negative selection by magnetic cell separation using a Midi MACS separator (Myltenyi Biotec). CD45 negative cell populations were then incubated with CD31 microbeads (Myltenyi Biotec) and positively selected by magnetic cell separation using the same separator. The obtained CD45–CD31+ cells were cultured as mouse pulmonary endothelial cells. Purity of the magnetically selected CD45–CD31+ endothelial cells was confirmed by flow cytometry and was greater than 90% (data not shown).

### 4.4. Cell Culture

Endothelial cells were cultured at a density of 2 × 10^6^ cells/well on 0.5% gelatin-coated, black-walled, clear-base plates in endothelial cell basal medium-2 supplemented with 5% fetal bovine serum and endothelial growth factors (EGM–2 MV bullet kit; Lonza, Walkersville, MD, USA) at 37 °C with 95% air, 5% CO_2_. After 3 days, non-adherent cells were removed and fresh medium was added. Subsequently, the medium was changed every other day. Cells were used for experiments 7 days after sorting. The conditioned medium collected from the cell cultures was stored at −80 °C for analysis of proteins. Lewis lung carcinoma cells (3LL; Japanese Collection of Research Bioresources cell bank, Osaka, Japan) were cultured at a density of 6 × 10^5^ cells/well in RPMI 1640 medium (Thermo Fisher Scientific, Tokyo, Japan) with 10% fetal bovine serum.

3LL cells and mouse pulmonary endothelial cells were cultured under both normoxic and hypoxic conditions [19,44,50]. Cells were cultured under normoxic conditions (20% O_2_) in a normal CO_2_ cell culture incubator (MCO-170AIC-PJ; PHC Holdings Corporation, Tokyo, Japan) or under hypoxic conditions (1% O_2_) in a multigas incubator (MCO-5M-PJ; PHC Holdings Corporation). Hypoxic conditions were created by filling the chamber with 1% O_2_, 5% CO_2_ and balanced N_2_. The oxygen concentration inside the incubator was continuously monitored with an internal zirconia sensor. After culture for 72 h under hypoxic conditions, cells were rapidly processed for RNA extraction or lysed for Western blot analysis.

### 4.5. Quantitative Real-Time PCR PCR (qRT-PCR)

For RNA isolation, endothelial cells and 3LL cells were lysed in RLT buffer (Qiagen, Valencia, CA, USA). Total RNA was then extracted using an RNeasy Mini Kit (Qiagen) with homogenization using a QIA shredder (Qiagen) according to the manufacturer’s instructions. The quality of total RNA samples was confirmed with a spectrophotometer (DeNovix DS–11; Scrum, Tokyo, Japan). Complementary DNA was generated from 1 μg RNA using a high-capacity cDNA reverse transcription kit (Applied Biosystems, Foster City, CA, USA) according to the manufacturer’s instructions. qRT-PCR was performed with the THUNDERBIRD SYBR qPCR mix (TOYOBO, Tokyo, Japan) and the Step One Plus system (Applied Biosystems). Relative quantification of target gene transcript levels was standardized to that of the β-actin gene and expressed as fold-change derived from ΔCt values, the Ct of a gene of interest minus the Ct of the β-actin gene from the same sample. Primers were from Thermo Fisher Scientific and primer sequences are shown in an additional file: Appendix A.

### 4.6. Quantification of Proteins

Proteins released from endothelial cells were quantified in samples of conditioned medium using ELISA kits for TGF-β1 (R&D Systems, Minneapolis, MN, USA), PDGF-B (R&D Systems), PDGF-C (Cloud-Clone Corp, Wuhan, China), PDGF-D (MyBiosource Inc., San Diego, CA, USA) or CTGF (Cusabio Life Science, Wuhan, China) according to the manufacturers’ protocols. Collagen content in conditioned medium was quantified using the Sircol Collagen Assay Kit (Biocolor Ltd., Carrickfergus, UK) according to the manufacturer’s protocol.

### 4.7. Western Blot Analysis

Cells were lysed in RIPA Buffer (Nacalai Tesque, Inc., Kyoto, Japan) containing a protease and phosphatase inhibitor cocktail (Fujifilm Wako Pure Chemical Corporation, Kyoto, Japan). Protein concentration was determined using a Pierce BCA protein assay kit (Thermo Fisher Scientific). Ten micrograms of protein were denatured in Laemmli sample buffer and loaded on 4–15% Mini-PROTEAN TGX Precast Gels (Bio-Rad Laboratories, Richmond, CA, USA). After electrophoretic separation, the proteins were transferred onto PVDF membranes (MilliporeSigma, Billerica, MA, USA). Membranes were blocked overnight at 4 °C with Tris-buffered saline (TBS) containing 5% non-fat dried milk or bovine serum albumin. Following blocking, membranes were incubated overnight at 4 °C with primary antibodies, including anti-HIF-1α (ab179483; 1:1000; Abcam, Cambridge, UK), anti-HIF-2α (ab109616; 1:1000; Abcam), and anti-β-actin (GTX109639; 1:1000; GeneTex, Irvine, CA, USA). Proteins were incubated with secondary antibodies (anti-rabbit or anti-mouse IgG antibody; Cat. 7074 and Cat. 7076, respectively; 1:20,000; Cell Signaling Technology, Danvers, MA, USA) for 1 h at room temperature in TBS containing 5% non-fat dried milk. The chemiluminescence signal was detected using enhanced chemiluminescent substrate (SuperSignal™ West Femto Trial Kit; Thermo Fisher Scientific).

### 4.8. Immunofluorescence Staining

For immunofluorescence staining, cells were fixed in 4% paraformaldehyde for 10 min, incubated in blocking solution (5% goat serum and 0.5% Triton X-100 in PBS) for 30 min and then incubated with unconjugated anti-α-SMA antibody (A5228; Sigma-Aldrich, St Louis, MO, USA), anti-HIF-2α antibody (ab109616; Abcam or NB100-132; Novus) or anti-CD31 antibody (ab28364; Abcam). Cells were then incubated with Alexa-Fluor 488 (green) conjugated anti-mouse IgG2a antibody (Invitrogen, Carlsbad, CA, USA), Alexa-Fluor 568 (red) conjugated anti-mouse IgG antibody (ab1500113; Abcam), or Alexa-Fluor 568 (red) conjugated anti-rabbit IgG antibody (ab175471; Abcam) and with Hoechst 33342 (Sigma-Aldrich) for 60 min. Cells were imaged and images captured with an IX83 microscope (Olympus, Tokyo, Japan) and the ratios of α-SMA- or HIF-2α-positive cells to total cell numbers estimated from Hoechst 33342 nuclear staining were calculated using Image-J (National Institutes of Health, Bethesda, MD, USA).

### 4.9. Statistical Analysis

All data are presented as the mean ± standard error of the mean. Differences between two groups were evaluated with the Mann–Whitney U-test. For multiple group comparisons, Tukey’s test was performed. A value of *p* < 0.05 was considered to be statistically significant. All statistical analyses were performed with EZR (Saitama Medical Center, Jichi Medical University, Saitama, Japan), which is a graphical user interface for R (The R Foundation for Statistical Computing, Vienna, Austria). More precisely, it is a modified version of the R commander designed to add statistical functions frequently used in biostatistics.

## Figures and Tables

**Figure 1 ijms-23-08996-f001:**
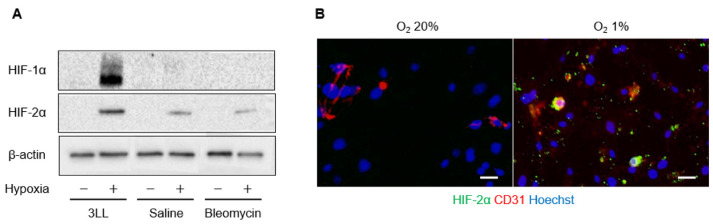
HIF-1 expression. (**A**) Western blot images of Lewis lung carcinoma (3LL) cells and endothelial cells from saline and bleomycin-treated lungs, cultured under normoxic (20%) and hypoxic (1%) conditions. Membranes were incubated with anti-HIF-1α, anti-HIF-2α, and anti-β-actin (loading control) primary antibodies. The chemiluminescence signal was detected using enhanced chemiluminescent substrate. (**B**) Representative fluorescence microscopy images of endothelial cells from bleomycin-treated lungs, cultured under normoxic (20%) and hypoxic (1%) conditions. The HIF-2α (green) and CD31 (red) stained images were merged with nuclear stained (Hoechst, blue) images. Scale bars indicate 50 μm.

**Figure 2 ijms-23-08996-f002:**
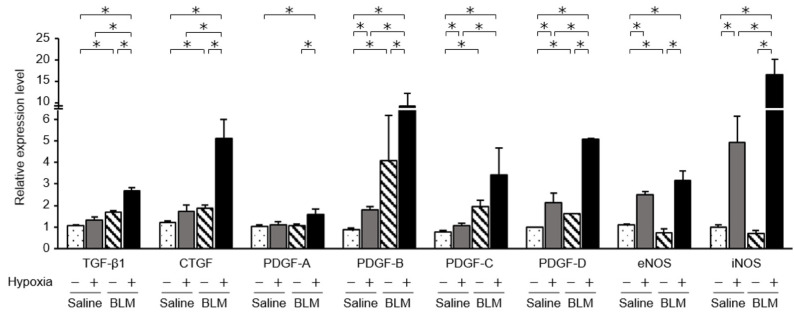
Expression of mediators, determined by quantitative real-time PCR. mRNA levels of various mediators were compared between endothelial cells from saline- and bleomycin-treated mouse lungs. Quantitative real-time PCR was performed using three independently prepared cDNA samples from endothelial cells harvested from saline- or bleomycin-treated lungs on day 7. Results were normalized to expression levels in endothelial cells from untreated lungs at day 0. Data are means ± standard error of the mean for three mice. * *p* < 0.05.

**Figure 3 ijms-23-08996-f003:**
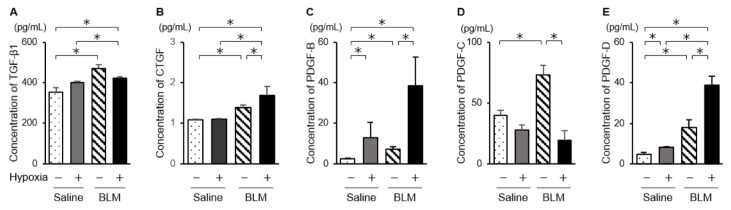
Fibrotic mediator proteins released from endothelial cells. Protein levels of TGF-β1 (**A**), CTGF (**B**), PDGF-B (**C**), PDGF-C (**D**), and PDGF-D (**E**) released from endothelial cells were quantified by ELISA. Data are means ± standard error of the mean for three mice. * *p* < 0.05.

**Figure 4 ijms-23-08996-f004:**
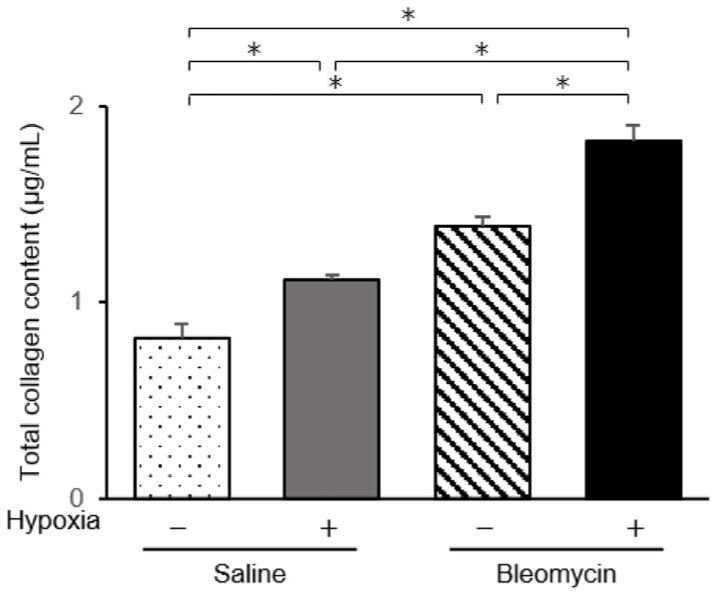
Total collagen content in endothelial cells. Total soluble collagen content in the culture medium from endothelial cells isolated from saline- and bleomycin-treated lungs was quantified by the Sircol assay. Results are means ± standard error of the mean from three mice per group. * *p* < 0.05.

**Figure 5 ijms-23-08996-f005:**
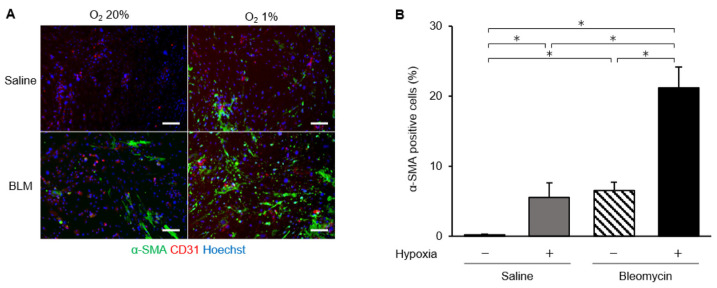
Hypoxia-induced changes of α-SMA staining in endothelial cells. (**A**) Representative fluorescence microscopy images of endothelial cells from saline- and bleomycin-treated lungs, cultured under normoxic (20%) and hypoxic (1%) conditions. The α-SMA (green) and CD31 (red) stained images were merged with those showing a nuclear stain (Hoechst, blue). Scale bars indicate 100 μm. (**B**) Comparison of the percentage of α-SMA-positive endothelial cells from saline- and bleomycin-treated lungs, cultured under normoxic (20%) and hypoxic (1%) conditions. The percentage of α-SMA-positive cells is expressed as the percentage of the total cell number, estimated from Hoechst 33342 nuclear staining. Data are means ± standard error from three mice. * *p* < 0.05.

**Figure 6 ijms-23-08996-f006:**
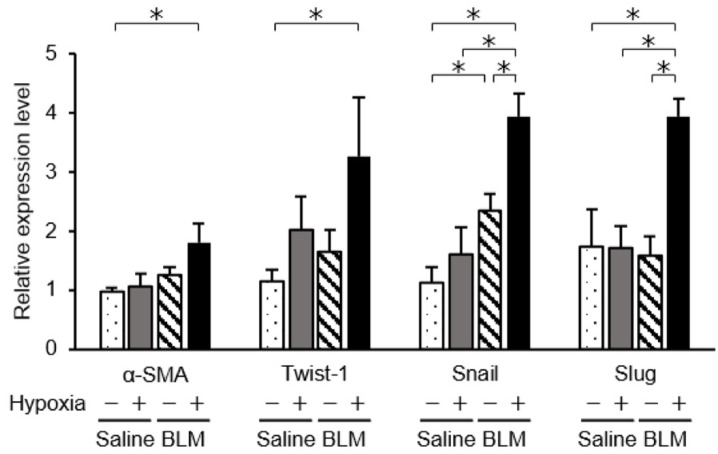
Expression of endothelial to mesenchymal transition markers determined by quantitative real-time PCR. mRNA levels of endothelial to mesenchymal transition markers were compared between endothelial cells from saline- and bleomycin-treated mouse lungs cultured under normoxic (20%) and hypoxic (1%) conditions. Quantitative real-time PCR was performed using three independently prepared cDNA samples from endothelial cells harvested from saline- or bleomycin-treated lungs on day 7. Results were normalized to expression levels in endothelial cells from untreated lungs at day 0. Data are means ± standard error of the mean for three mice. * *p* < 0.05.

**Figure 7 ijms-23-08996-f007:**
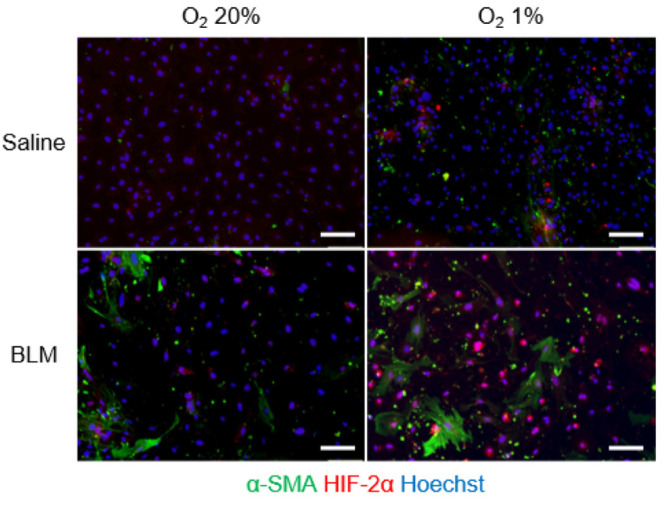
Hypoxia-induced co-expression of HIF-2α and α-SMA in endothelial cells. Representative fluorescence microscopy images of endothelial cells from saline- and bleomycin-treated lungs, cultured under normoxic (20%) and hypoxic (1%) conditions. The HIF-2α (red) and α-SMA (green) stained images were merged with those showing a nuclear stain (Hoechst, blue). Scale bars indicate 100 μm.

**Table 1 ijms-23-08996-t001:** Relative mRNA levels associated with endothelial cell damage.

	Saline		Bleomycin	
	20% O_2_	1% O_2_	20% O_2_	1% O_2_
vWF	1.33 ± 0.16	8.79 ± 1.55 *	3.22 ± 0.57 *	17.5 ± 0.98 *
MMP-12	1.37 ± 0.20	78.2 ± 5.96 *	1.98 ± 0.13	906 ± 148 *
PAI-1	1.08 ± 0.26	14.4 ± 1.00 *	3.02 ± 0.21 *	16.1 ± 2.75 *

The table shows expression of endothelial damage markers, von Willebrand factor (vWF), matrix metalloproteinase (MMP)-12, and plasminogen activator inhibitor 1 (PAI-1) in pulmonary endothelial cells isolated from saline-treated and bleomycin-treated mice. mRNA levels were determined by quantitative real-time PCR. Total RNA was isolated from magnetically sorted CD45−CD31+ mouse pulmonary endothelial cells at 7 days after saline or bleomycin intratracheal injection. PCR was performed using three independently prepared cDNA samples from endothelial cells. Results were normalized to expression levels in endothelial cells from untreated lungs at day 0 and relative quantification of target gene transcript levels was standardized to those of the β-actin gene and are expressed as fold-changes derived from ΔCt values. Data are means ± standard error from three mice. * *p* < 0.01, compared with endothelial cells from saline-treated mice cultured in normoxic conditions (20% O_2_).

## Data Availability

The datasets used and analyzed during the current study are available from the corresponding author upon reasonable request.

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
