# Peer review of "Effect of Hypoxia on Pulmonary Endothelial Cells from Bleomycin-Induced Pulmonary Fibrosis Model Mice"

_ijms, 2022, doi:10.3390/ijms23168996_

Round 1

Reviewer 1 Report

The manuscript proposed for publication is a descriptive image of the behavior of the cells from bleomycin-treated mice under hypoxic conditions. The cells, already described to be a fibrotic model, were investigated at low oxygen concentrations, with the aim to clarify if they undergo endothelial mesenchymal transition (EndMT) in line 87.

The paper was difficult to evaluate since all the symbols in the plots were lost. Moreover, because of the lack of symbols and description in figure 4 caption, we are not even sure that hypoxic cells produce more collagen and therefore are more fibrotic.

Although the manuscript has no evident scientific flaws and could be corrected after a big work of revision, I am not sure it adds such important information about fibrosis to be published in this form. There are some hints about the higher susceptibility of ECs derived from bleomycin-treated lungs, but no investigations into the effect of hypoxia on the development of fibrosis. That hypoxic conditions increase HIF-1 is well known, and the only data added would be that fibrotic-prone cells react more quickly and with higher effects. Nevertheless, there are no experiments, investigation, or speculation (even in the three pages long discussions) about the mechanism or the connection between HIF and “accumulation of excess extracellular matrix” reported in the introduction and not proved. The EndMT process (see 10.1152/physrev.00021.2018 as reference) is a complex process, but the authors do not address any of the mechanisms apart from the expression of alfa-SMA, not even the type of collagen produced.

Suggestions:

All the symbols in the plots are lost.  Because of the lack of symbols and description in figure 4 caption, we are not even sure that hypoxic cells produce more collagen and therefore are more fibrotic.

Microscope images have no error bars.

Figure 1: in text lane 103 add “normoxic and” before hypoxic. Figure 1A does not correspond to the description in the first paragraph of 2.2: the image loaded has no signal for any of the target proteins: was it a problem with visualization?

Why in 3LL cells is there no signal of beta-actin? Or any other reference protein? The western blot for those cells did not give any information about HIS protein if there is no reference protein to show us that there was something loaded.

Paragraph 2.3: line 131, PAI-1 does not seem different in hypoxic conditions of the two samples.

Paragraph 2.4: the production of collagen was indeed quite important to assess the fibrotic condition of the cells, but it is isolated, not well connected with other data and there is no gene expression as reported for all the other protein targets.

Line 195 the sentence is ambiguous, we can assume that hypoxic ECs have higher alfa-SMA with respect to normoxia, but the text does not report anything.

Figure 6 in the text was not well described: slug? Twist? And moreover, what meaning do they have? Since they are involved in the EndMT transition, they should be better investigated.

Captions of figures 5 and 6 contain elements that should be moved to materials and methods.

The discussion is loo long and not well focused on the real results reported in the manuscript.

Author Response

Response to Reviewer 1

Reviewer 2 Report

In this manuscript, Akahori and colleagues showed original data about the impact of hypoxia on pulmonary endothelial cells during pulmonary fibrogenesis demonstrating that exposure to hypoxia damaged endothelial cells and enhanced fibrogenesis-related damage in bleomycin-treated pulmonary endothelial cells.

This in an interesting work providing new insights of how endothelial cells isolated from a pathological context (here bleomycin-induced pulmonary fibrosis) react differently to a cellular stress than cells from normal lungs.

Here are some comments to improve this manuscript:

Major points:

In figure 1 the authors didn’t see any expression of HIF1 alpha in ECs during hypoxia.  While I agree that the kinetics of HIF1 and HIF2 are not the same, this is still a little bit unconvincing and I am wondering if the cells are really in hypoxia in this experiment. To check this, I suggest that the authors at least perform some qPCR targeting some known HIF1 targets such as VEGFA, GLUT1, PGK and Hexokinase.  They could also perform Western Blot analysis on eIF2α phosphorylation as eIF2α phosphorylation downregulates protein synthesis in various stress conditions, including hypoxia (Koumenis and Wouters, 2006).

In line 135, the authors wrote that “pre-stimulation with bleomycin enhanced hypoxia-induced endothelial cell injury.” It would be very interesting to assess if this pre-stimulation works specifically with hypoxia or if it has a global effect on cellular stresses. To answer this question, I would suggest to test for example endoplasmic reticulum stress with tunicamycin treatment and perform the same qPCR experiments as in Figure 2 and 6 on ECs from bleomycin-treated or control lungs.

The authors did not discuss the differences in results between Figure 2 and 3. How can they explain that the increase of TGFβ mRNA induced by hypoxia on bleomycin-treated ECs is no longer detectable at the protein level? Same goes for CTGF that seems to be much more induced at the mRNA level than at the protein level

In figure 5, the number of αSMA positive cells strongly increase in hypoxia in bleomycin-treated ECs. Yet, in figure 6 this increase is not detected at the RNA level: can the authors comment on that?

Minor points:

The markers used for the stainings in Figure 5A should be written on the pictures directly in the Figure to allow readers to understand what has be done in the figure without needing to go back to the text.

The authors claimed that “Bleomycin-stimulation might represent a priming state” (line 247) able to sensitize the cells to cellular stresses like hypoxia.  While this concept is very interesting, the authors did not discuss about what could be this “priming signal” in the pulmonary fibrosis: crosstalk between EC and inflammatory cells? Crosstalk between EC and the fibrotic extracellular matrix? This part could lead to interesting hypothesis and it is lacking in the discussion part of the manuscript

Round 2

Reviewer 2 Report

the authors have adressed all of my concerns